# On the Frequency-Bias of Coordinate-MLPs

**Sameera Ramasinghe**            **Lachlan Macdonald**            **Simon Lucey**

{firstname.lastname}@adelaide.edu.au
University of Adelaide

## Abstract

We show that typical implicit regularization assumptions for deep neural networks (for regression) do not hold for coordinate-MLPs, a family of MLPs that are now ubiquitous in computer vision for representing high-frequency signals. Lack of such implicit bias disrupts smooth interpolations between training samples, and hampers generalizing across signal regions with different spectra. We investigate this behavior through a Fourier lens and uncover that as the bandwidth of a coordinate-MLP is enhanced, lower frequencies tend to get suppressed unless a suitable prior is provided explicitly. Based on these insights, we propose a simple regularization technique that can mitigate the above problem, which can be incorporated into existing networks without any architectural modifications.

## 1 Introduction

It is well-established that deep neural networks (DNN), despite mostly being used in the over-parameterized regime, exhibit remarkable generalization properties without explicit regularization Neyshabur et al. [2014], Zhang et al. [2017], Savarese et al. [2019], Goodfellow et al. [2016]. This behavior questions the classical theories that predict an inverse relationship between model complexity and generalization, and is often referred to in the literature as the "implicit regularization" (or implicit bias) of DNNs Li and Liang [2018], Kubo et al. [2019], Soudry et al. [2018], Poggio et al. [2018]. Characterizing this surprising phenomenon has been the goal of extensive research in recent years.

In contrast to this mainstream understanding, we show that coordinate-MLPs (or implicit neural networks), a class of MLPs that are specifically designed to overcome the *spectral bias* of regular MLPs, do not follow the same behavior. Spectral bias, as the name suggests, refers to the propensity of DNNs to learn functions with low frequencies, making them unsuited for encoding signals with high frequency content. Coordinate-MLPs, on the other hand, are architecturally modified MLPs (via specific activation functions Sitzmann et al. [2020], Ramasinghe and Lucey [2021] or positional embedding schemes Mildenhall et al. [2020], Zheng et al. [2021]) that can learn functions with high-frequency components. By virtue of this unique ability, coordinate-MLPs are now being used extensively in computer vision tasks for representing signals including texture generation Henzler et al. [2020], Oechsle et al. [2019], Henzler et al. [2020], Xiang et al. [2021], shape representation Chen and Zhang [2019], Deng et al. [2020], Tiwari et al. [2021], Genova et al. [2020], Basher et al. [2021], Mu et al. [2021], Park et al. [2019], and novel view synthesis Mildenhall et al. [2020], Niemeyer et al. [2020], Saito et al. [2019], Sitzmann et al. [2019], Yu et al. [2021], Pumarola et al. [2021], Rebain et al. [2021], Martin-Brualla et al. [2021], Wang et al. [2021], Park et al. [2021]. However, as we will show in this paper, these architectural alterations entail an unanticipated drawback: coordinate-MLPs, trained by conventional means via stochastic gradient descent (SGD), are incapable of simultaneously generalizing well at both lower and higher ends of the spectrum, and thus, are not automatically biased towards less complex solutions (Fig. 1). Based on this observation, we question the popular understanding that the implicit bias of neural networks is more tied to SGD than the architecture Zhang et al. [2017, 2021].

36th Conference on Neural Information Processing Systems (NeurIPS 2022).

Strictly speaking, the term "generalization" is not meaningful without context. For instance, consider a regression problem where the training points are sparsely sampled. Given more trainable parameters than the number of training points, a neural network can, in theory, learn infinitely many interpolants of the training points. Therefore, the challenge is to learn a function within a space restricted by certain priors and intuitions regarding the problem at hand. The generalization then can be measured by the extent to which the learned function is close to these prior assumptions about the task. Within a regression problem, one intuitive solution that is widely accepted by the practitioners (at least from an engineering perspective) is to have a form of "smooth" interpolation between the training points, where the low-order derivatives are bounded Bishop [2007]. In classical machine learning, in order to restrict the class of learned functions, explicit regularization techniques were used Craven and Wahba [1978], Wahba [1975], Kimeldorf and Wahba [1970]. Paradoxically, however, over-parameterized neural networks with extremely high capacity prefer to converge to such smooth solutions without any explicit regularization, despite having the ability to fit more complex functions Zhang et al. [2017, 2021].

Although this expectation of smooth interpolation is valid for both *a*) general regression problems with regular MLPs and *b*) high-frequency signal encoding with coordinate-MLPs, a key difference exists between their end-goals. In a general regression problem, we do not expect an MLP to perfectly fit the training data. Instead, we expect the MLP to learn a smooth curve that achieves a good trade-off between the bias and variance (generally tied to the anticipation of noisy data), which might not exactly overlap with the training points. In contrast, coordinate-MLPs are particularly expected to *perfectly fit* the training data that may include both low and high fluctuations, while interpolating smoothly between samples. This difficult task requires coordinate-MLPs to preserve a rich spectrum with both low and high frequencies (often with higher spectral energy for low frequencies, as the power spectrum of natural signals such as images tends to behave as $1/f^2$ Ruderman [1994]). We show that coordinate-MLPs naturally do *not* tend to converge to such solutions, despite the existence of possible solutions within the parameter space.

It should be pointed out that it *has* indeed been previously observed that coordinate-MLPs tend to produce noisy solutions when the bandwidth is increased excessively Tancik et al. [2020], Ramasinghe and Lucey [2021], Sitzmann et al. [2020]. However, our work complements these previous observations: *First*, the existing works do not offer an explanation on why providing a coordinate-MLP with the capacity to add high frequencies to the spectrum would necessarily affect lower frequencies. In contrast, we elucidate this behavior through a Fourier lens, and show that as the spectrum of the coordinate-MLPs is enhanced via hyper-parameters or depth, the lower frequencies tend to be suppressed (for a given set of weights), hampering their ability to interpolate smoothly within low-frequency regions. *Second*, we interpret this behaviour from a model complexity angle, which allows us to incorporate network-depth into our analysis. *Third*, we show that this effect is common to many types of coordinate-MLPs, *i.e.,* MLPs with *a*) positional embeddings Mildenhall et al. [2020], *b*) periodic activations Sitzmann et al. [2020], and *c*) non-periodic activations Ramasinghe and Lucey [2021]. Finally, we propose a simple regularization term that can enforce coordinate-MLPs to preserve both low and high frequencies in practice, enabling better generalization across the spectrum.

Our contributions are summarized below:

- We show that coordinate-MLPs are not implicitly biased towards low-complexity solutions, *i.e.,* typical implicit regularization assumptions do not hold for coordinate-MLPs. A bulk of previous mainstream works try to connect the implicit bias of neural networks to the properties of the optimization procedure (SGD), rather than the architecture Zhang et al. [2017, 2021]. On the contrary, we provide counter-evidence that the implicit bias of neural networks might indeed be strongly tied to the architecture.

- We present a general result (Theorem 4.1), that can be used to obtain the Fourier transform of shallow networks with arbitrary activation functions and multi-dimensional inputs, even when the Fourier expressions are not directly integrable over the input space. Utilizing this result we derive explicit formulae to study shallow coordinate-MLPs from a Fourier perspective.

- Using the above expressions, we show that shallow coordinate-MLPs tend to suppress lower frequencies when the bandwidth is increased via hyper-parameters or depth, disrupting smooth interpolations. Further, we empirically demonstrate that these theoretical insights from shallow networks extrapolate well to deeper ones.

- We propose a simple regularization technique to enforce smooth interpolations between training samples while perfectly fitting the training points, preserving a rich spectrum. The proposed technique can be easily applied to existing coordinate-MLPs without any architectural modifications.

Our analysis stands out from previous theoretical research Tancik et al. [2020], Ramasinghe and Lucey [2021], Zheng et al. [2021] on coordinate-MLPs for several reasons: *a*) we work in a relatively realistic setting excluding assumptions such as infinite-width networks or linear models. Although we *do* consider shallow networks, we also take into account the effect of increasing depth. *b*) All our expressions are derived from first principles and do not demand the support of additional literature such as neural tangent kernel (NTK) theory. *c*) we analyze different types of coordinate-MLPs within a single framework where the gathered insights are common across each type.

## 2  Related works

Understanding and characterizing the astonishing implicit generalization properties of DNNs has been an important research topic in recent years. Despite the overwhelming empirical evidence, establishing a rigorous theoretical underpinning for this behavior has been a challenge to this date. The related research can be broadly categorized into two: investigating implicit regularization on the *a*) weight space Bishop [1995], Soudry et al. [2018], Poggio et al. [2018], Gidel et al. [2019] and the *b*) function space Maennel et al. [2018], Kubo et al. [2019], Heiss et al. [2019]. Notably, Maennel et al. [2018] analyzed ReLU networks under macroscopic assumptions and affirmed that for given input data, there are only finitely many, "simple" functions that can be obtained using regular ReLU MLPs, independent of the network size. Kubo et al. [2019] showed that ReLU-MLPs interpolate between training samples almost linearly, and Heiss et al. [2019] took a step further, proving that the interpolations

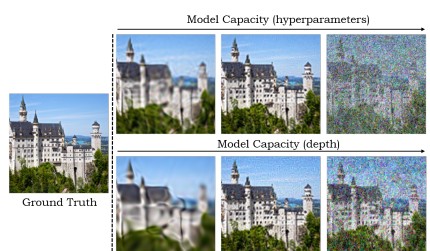

Figure 1: **Implicit regularization assumptions do not hold for Coordinate-MLPs**. The network is trained with 10% of the pixels in each instance. As the capacity of the network is increased via depth or hyperparameters, coordinate-MLPs tend to produce more complex solutions even though they are trained with SGD. This is in contrast to the regular-MLPs, where the networks converge to "smooth" solutions independent of the model capacity Kubo et al. [2019], Heiss et al. [2019]. Further, this provides evidence that the implicit regularization of neural networks might be strongly linked to the architecture rather than the optimization procedure, as opposed to a mainstream understanding Zhang et al. [2017, 2021].

can converge to (nearly) spline approximations. Seminal works by Zhang et al. [2017] and Zhang et al. [2021] showed that these generalization properties are strongly connected to the SGD. In contrast, we show that coordinate-MLPs tend to converge to complex solutions (given enough capacity), despite being trained with SGD. Further, all above works consider shallow networks in order to obtain precise theoretical guarantees. Similarly, we also utilize shallow architectures for a part of our analysis. However, our work differs from some of the above-mentioned theoretical work, as we do not focus on establishing rigorous theoretical bounds on generalization. Rather, we uncover, to the best of our knowledge, a critically overlooked shortcoming of coordinate-MLPs, and present plausible reasoning for this phenomenon from a Fourier perspective. We give a brief exposition of coordinate-MLPs in the next section.

## 3  Coordinate-MLPs

Coordinate-MLPs aim to encode continuous signals $f : \mathbb{R}^n \to \mathbb{R}^m$, *e.g.,* images, sound waves, or videos, as their weights. The inputs to the network typically are low-dimensional coordinates, *e.g.,* $(x, y)$ positions, and the outputs are the sampled signal values at each coordinate *e.g.*, pixel intensities. The key difference between coordinate-MLPs and regular MLPs is that the former is designed to encode signals with higher frequencies – mitigating the spectral bias of the latter – via specific architectural modifications. Below, we will succinctly discuss three types of coordinate-MLPs.

**Random Fourier Feature (RFF) MLPs** are compositions of a positional embedding layer and subsequent ReLU layers. Let $\Omega$ denote the probability space $\mathbb{R}^n$ equipped with the Gaussian measure

of standard deviation $2\pi\sigma > 0$. For $D \geq 1$, write elements $\mathbf{L}$ of $\Omega^n$ as matrices $[\mathbf{l}_1, \ldots, \mathbf{l}_D]$ composed of $D$ vectors drawn independently from $\Omega$. Then, the random Fourier feature (RFF) positional embedding $\gamma : \Omega^D \times \mathbb{R}^n \to \mathbb{R}^{2D}$ is

$$\gamma(\mathbf{L}, \mathbf{x}) := [\mathbf{e}(\mathbf{l}_1, \mathbf{x}), \ldots, \mathbf{e}(\mathbf{l}_D, \mathbf{x})]^T, \tag{1}$$

where $\mathbf{e} : \Omega \times \mathbb{R}^n \to \mathbb{R}^2$ is the random function defined by the formula

$$\mathbf{e}(\mathbf{l}, \mathbf{x}) := [\sin(\mathbf{l} \cdot \mathbf{x}), \cos(\mathbf{l} \cdot \mathbf{x})]. \tag{2}$$

The above layer is then followed by a stack of ReLU layers.

**Sinusoidal MLPs** were originally proposed by Sitzmann et al. [2020]. Let $\mathbf{W}$ and $\mathbf{b}$ be the weights and the bias of a hidden layer of a sinusoidal coordinate-MLP, respectively. Then, the output of the hidden-layer is $\sin(2\pi a(\mathbf{W} \cdot \mathbf{x} + \mathbf{b}))$, where $a$ is a hyper-parameter that can control the spectral-bias of the network and $\mathbf{x}$ is the input.

**Gaussian MLPs** are a recently proposed type of coordinate-MLP by Ramasinghe and Lucey [2021]. The output of a Gaussian hidden-layer is defined by $e^{-(\frac{(\mathbf{W} \cdot \mathbf{x} + \mathbf{b})^2}{2\sigma^2})}$, where $\sigma$ is a hyper-parameter.

In the next section, we will derive a general result that can be used to obtain the Fourier transform of an arbitrary multi-dimensional shallow MLP (given that the 1D Fourier transform of the activation function exists), which is used as the bedrock in a bulk of our derivations later.

## 4 Fourier transform of a shallow MLP

Let $\mathcal{G} : \mathbb{R}^n \to \mathbb{R}$ be an MLP with a single hidden layer with $m$ neurons, and a point-wise activation function $\alpha : \mathbb{R} \to \mathbb{R}$. Suppose we are interested in obtaining the Fourier transform $\hat{\mathcal{G}}$ of $\mathcal{G}$. Since the bias only contributes to the DC component, we formulate $\mathcal{G}$ as $\mathcal{G} = \sum_{i=1}^m w_i^{(2)} \alpha(\mathbf{w}_i^{(1)} \cdot \mathbf{x})$, where $\mathbf{w}_i^{(1)}$ are rows of the fist-layer affine weight matrix, input to the $i^{th}$ hidden neuron is $\mathbf{w}_i^{(1)} \cdot \mathbf{x}$, and $w_i^{(2)}$ are the weights of the last layer. Since the Fourier transform is a linear operation, it is straightforward to see that

$$\hat{\mathcal{G}} = \sum_{i=1}^m w_i^{(2)} \hat{\alpha}(\mathbf{w}_i^{(1)} \cdot \mathbf{x}). \tag{3}$$

Thus, by obtaining the Fourier transform of $f(\mathbf{x}) := \alpha(\mathbf{w} \cdot \mathbf{x})$, it is possible to derive the Fourier transform of the MLP. However, doing so using the standard definition of the multi-dimensional Fourier transform can be infeasible in some cases. For example, consider a Gaussian-MLP. Then, one can hope to calculate the Fourier transform of $f(\mathbf{x})$ as $\hat{f}(\mathbf{k}) = \int_{\mathbb{R}^n} e^{-(\frac{(\mathbf{w} \cdot \mathbf{x})^2}{2\sigma^2} + 2\pi i \mathbf{x} \cdot \mathbf{k})} d\mathbf{x}$. However, note that there exists an $n - 1$ dimensional subspace where $(\mathbf{w} \cdot \mathbf{x})$ is zero and thus, the first term inside the exponential becomes $0$ in these cases. Therefore the Fourier transform does not exist as a function. Nonetheless, $f$ defines a tempered distribution, and its Fourier transform $\hat{f}$ can therefore be calculated by fixing a Schwartz test function $\varphi$ on $\mathbb{R}^n$ and using the identity $\langle \hat{f}, \varphi \rangle = \langle f, \hat{\varphi} \rangle$, defining the Fourier transform of the tempered distribution $f$. Here $\langle \cdot, \cdot \rangle$ denotes the pairing between distributions and test functions (see Chapter 8 of Friedlander and Joshi [1999] for this background). Formally, we present the following theorem, whose proof we defer to the appendix.

**Theorem 4.1.** *The Fourier transform of $f(\mathbf{x}) := \alpha(\mathbf{w} \cdot \mathbf{x})$ where $\mathbf{w}, \mathbf{x} \in \mathbb{R}^n$ is the distribution*

$$\hat{f}(\mathbf{k}) = \frac{(2\pi)^{\frac{n}{2}}}{|\mathbf{w}|} \hat{\alpha}\left(\frac{\mathbf{w}}{|\mathbf{w}|^2} \cdot \mathbf{k}\right) \delta_{\mathbf{w}}(\mathbf{k}),$$

*where $\delta_{\mathbf{w}}(\mathbf{k})$ is the Dirac delta distribution which concentrates along the line spanned by $\mathbf{w}$.* $\qquad \square$

Next, using the above result, we will gather useful insights into different types of coordinate-MLPs from a Fourier perspective.

# 5 Effect of hyperparameters

In this section, we will primarily explore the effect of hyperparameters on the spectrum of coordinate-MLPs, among other key insights.

## 5.1 Gaussian-MLP

The Fourier transform of the Gaussian activation is $\hat{\alpha}(k) = \sqrt{2\pi}\sigma e^{-(\sqrt{2}\pi k\sigma)^2}$. Now, using Theorem 4.1, we can obtain the Fourier transform of a Gaussian-MLP with a single hidden layer as

$$\sum_{i=1}^{m} w_i^{(2)} \frac{(2\pi)^{\frac{n+1}{2}}\sigma}{|\mathbf{w}_i^{(1)}|} e^{-\left(\sqrt{2}\pi \frac{\mathbf{w}_i^{(1)}}{|\mathbf{w}_i^{(1)}|^2} \cdot \mathbf{k}\sigma\right)^2} \delta_{\mathbf{w}_i^{(1)}}(\mathbf{k}). \tag{4}$$

**Discussion:** For fixed $\mathbf{w}_i^{(1)}$'s and $\sigma$, the spectral energy is decayed as $\mathbf{k}$ is increased. A practitioner can increase the energy of higher frequencies by decreasing $\sigma$. Second, although decreasing $\sigma$ can increase the energy of higher frequencies (for given weights), it simultaneously suppresses the energies of lower frequencies due to the $\sigma$ term outside the exponential. Third, one can achieve the best of both worlds (*i.e.,* incorporate higher-frequencies to the spectrum while maintaining higher energies for lower-frequencies) by appropriately tuning $\mathbf{w}_i^{(1)}$'s. To make the above theoretical observations clearer, we demonstrate a toy example in Fig. 2.

## 5.2 Sinusoidal-MLPs

Similar to the Gaussian-MLP, the Fourier transform of a sinusoidal-MLP with a single hidden-layer can be obtained as

$$\sum_{i=1}^{m} w_i^{(2)} \frac{(2\pi)^{\frac{n}{2}}}{2|\mathbf{w}_i^{(1)}|} \delta_{\mathbf{w}_i^{(1)}}(\mathbf{k})\left(\delta\left(\frac{\mathbf{w}_i^{(1)}}{|\mathbf{w}_i^{(1)}|^2} \cdot \mathbf{k} - a\right) + \delta\left(\frac{\mathbf{w}_i^{(1)}}{|\mathbf{w}_i^{(1)}|^2} \cdot \mathbf{k} + a\right)\right), \tag{5}$$

**Discussion:** According to Eq. 5, the only frequencies present in the spectrum are multiples $\mathbf{k} = a\mathbf{w}_i^{(1)}$ of the weight vectors $\mathbf{w}_i^{(1)}$. It follows that the magnitudes of the frequencies present in the spectrum are lower-bounded by the number $a \min\{|\mathbf{w}_i|\}$. Thus for a given set of weights, one can add higher frequencies to the spectrum by increasing $a$. At the same time, the spectrum can be *balanced* by minimizing some $\mathbf{w}_i$'s accordingly.

## 5.3 RFF-MLPs

We consider a shallow RFF-MLP $\mathbb{R}^n \rightarrow \mathbb{R}$ with a single hidden ReLU layer. Recall that the positional embedding scheme is defined following Eq. 1 and 2. The positional embedding layer $\gamma$ is then followed by a linear transformation $A : \mathbb{R}^{2D} \rightarrow \mathbb{R}^{2D}$ yielding

$$f_2(\mathbf{L}, \mathbf{x})^i = \sum_j a_j^i \gamma(\mathbf{L}, \mathbf{x})^j,$$

which is then followed by ReLU activations and an affine transformation. ReLU activations can be approximated via linear combinations of various polynomial basis functions Ali et al. [2020], Telgarsky [2017]. One such set of polynomials is Newman polynomials defined as

$$N_m(x) := \prod_{i=1}^{m-1} (x + \exp(-i/\sqrt{n})).$$

It is important to note that although we stick to the Newman polynomials in the subsequent derivations, the obtained insights are valid for *any* polynomial approximation, as they only depend on the power terms of the polynomial approximation. Moving on, for $m \geq 5$, we show in the appendix that the spectrum of the RFF-MLP is concentrated on frequencies

$$\mathbf{k} = \sum_{j=1}^{D} \left((2a_{2j-1} - k_{2j-1}) + (2a_{2j} - k_{2j})\right)\mathbf{l}_j, \tag{6}$$

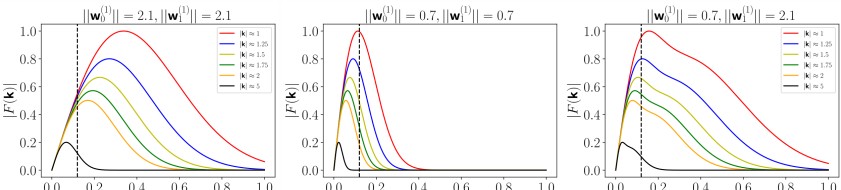

Figure 2: **Toy example (a Gaussian-MLP $\mathbb{R}^2 \to \mathbb{R}$ with a single hidden layer consisting of two neurons):**
The behavior of the spectrum against $\sigma$ and the weights of the network is shown. $\mathbf{w}_0^{(1)}$ and $\mathbf{w}_1^{(1)}$ are the first and second rows of the first-layer affine weight matrix. *Left:* By decreasing $\sigma$, the network can include higher frequencies to the spectrum. However, as the high-frequency components are added to the spectrum, the relative energies of the low-frequency components decrease. *Middle:* The network can still gain high energies for lower frequencies at lower $\sigma$ by decreasing $|\mathbf{w}_0^{(1)}|$ and $|\mathbf{w}_1^{(1)}|$. However, in this case, the high-frequency components are removed from the spectrum. *Right:* By appropriately tuning $|\mathbf{w}_0^{(1)}|$ and $|\mathbf{w}_1^{(1)}|$, the spectrum can include higher-frequency components while preserving the low-frequency energies. All the intensities are normalized for better comprehension.

for non-negative integers $k_1, \ldots, k_{2D}$ adding to $m - 2$, and for all non-negative integers $a_j \leq k_j$ for all $j = 1, \ldots, 2D$. Here we recall that $D$ is the dimension determined by the choice of RFF positional embedding.

**Discussion:** Recall that if $U, V \sim \mathcal{N}(0, \sigma^2)$, then, $Var(t_1 U + t_2 V) = (t_1^2 + t_2^2)\sigma^2$. Therefore the overall spectrum can be made wider by increasing the standard deviation $\sigma$ of the distribution from which the $\mathbf{l}_j$ are drawn. However, since the $D$ is constant, a larger $\sigma$ makes frequencies less concentrated in the lower end, reducing the overall energy of the low frequency components. Nonetheless, the spectrum can be altered via adjusting the network weights.

Thus far, we established that in all three types of coordinate-MLPs, a rich spectrum consisting of both low and high frequencies can be preserved by properly tuning the weights and hyperparameters. However, coordinate-MLPs tend to suppress lower frequencies when trained without explicit regularization (see Fig. 5). Moreover, note that in each Fourier expression, the Fourier components cease to exist if $\mathbf{k}$ is not in the direction of $\mathbf{w}_i^{(1)}$, due to the Dirac-delta term. Therefore, the angles between $\mathbf{w}_i^{(1)}$'s should be increased, in order to add frequency components in different directions.

# 6 Effect of depth

This section investigates the effects of increasing depth on the spectrum of coordinate-MLPs. Comprehensive derivations of the expressions in this section can be found in the Appendix. Let a stack of layers be denoted as $\kappa : \mathbb{R}^n \to \mathbb{R}^d$. Suppose we add another stack of layers $\eta : \mathbb{R}^d \to \mathbb{R}$ on top of $\kappa(\cdot)$ to construct the MLP $\eta \circ \kappa : \mathbb{R}^n \to \mathbb{R}$. The Fourier transform of the composite function then becomes,

$$\widehat{(\eta \circ \kappa)}(\mathbf{k}) = \left(\frac{1}{\sqrt{2\pi}}\right)^n \langle \hat{\eta}(\cdot), \beta(\mathbf{k}, \cdot) \rangle, \tag{7}$$

where, $\beta(\mathbf{k}, \mathbf{t}) = \int_{\mathbb{R}^n} e^{i(\mathbf{t} \cdot \kappa(\mathbf{x}) - \mathbf{k} \cdot \mathbf{x})}$. That is, the composite Fourier transform is the projection of $\hat{\eta}$ on to $\beta(\cdot)$. The maximum magnitude frequency $\mathbf{k}^*_{\eta \circ \kappa}$ present in the spectrum of $\widehat{(\eta \circ \kappa)}$ is

$$\mathbf{k}^*_{\eta \circ \kappa} = \max_{|\mathbf{u}|=1} (\mathbf{k}^*_{\hat{\eta}, \mathbf{u}} \max_{\mathbf{x}} (\mathbf{u} \cdot \kappa'(\mathbf{x}))), \tag{8}$$

where $\mathbf{k}_{\hat{\eta}, \mathbf{u}}$ is the maximum frequency of $\hat{\eta}$ along $\mathbf{u}$ Bergner et al. [2006]. Thus, the addition of a set of layers $\eta$ with sufficiently rich spectrum tends to increase the maximum magnitude frequency expressible by the network. Next, we focus on the impact on lower frequencies by such stacking. It is important to note that the remainder of the discussion in this section is not a rigorous theoretical derivation, but rather, an intuitive explanation.

We consider the 1D case for simplicity. Our intention is to explore the effect of adding more layers on $\langle \hat{\eta}(\cdot), \beta(\mathbf{k}, \cdot) \rangle$. Note that $\beta$ is an integral of an oscillating function with unit magnitude. The integral of an oscillating function is non-negligible only at points where the phase is close to zero, *i.e.,* points of stationary phase. It can be deduced that at such points the following condition needs to be satisfied,

$$\min(\kappa'(x)) < \frac{k}{t} < \max(\kappa'(x)). \tag{9}$$

From the previous analysis, we established that progressively adding layers increases the maximum frequency of the composite function. This causes $\kappa$ to have rapid fluctuations, encouraging $\min(\kappa'(x))$ to increase as the network gets deeper (note that the definition of $\kappa$ keeps changing as more layers are added). Note that as $\min(\kappa')$ is increased, for smaller $k$'s, Eq. 9 is not satisfied. On the other hand, $\beta$ is non-negligible only at points where Eq. 9 is satisfied. Thus, according to Eq. 9, smaller $k$'s causes the quantity $\langle \hat{\eta}(\cdot), \beta(\mathbf{k}, \cdot) \rangle$ to be smaller, which encourages suppressing the energy of the lower frequencies of $\widehat{(\eta \circ \kappa)}$. Rigorously speaking, increasing the bandwidth of the network by adding more layers does not necessarily have to increase $\min(\kappa'(x))$. However, our experimental results strongly suggest that this is the case. Summarizing our insights from Sec. 4, 5 and Sec. 6, we state the following remarks.

**Remark 6.1.** *In order to gain a richer spectrum, the columns of the affine weight matrices of the coordinate- MLPs need to point in different directions. Ideally, the column vectors should lie along the frequency directions in the target signal.*

**Remark 6.2.** *The spectrum of the coordinate-MLPs can be altered either by tuning the hyperparameters or changing the depth. In practice, when higher frequencies are added to the spectrum, the energy of lower frequencies tend to be suppressed, disrupting smooth interpolations between samples. However, better solutions exists in the parameter space, thus, an explicit prior is needed to bias the network weights towards those solutions.*

In the next section, we will focus on proposing a regularization mechanism that can aid the coordinate-MLPs in finding a solution with a better spectrum.

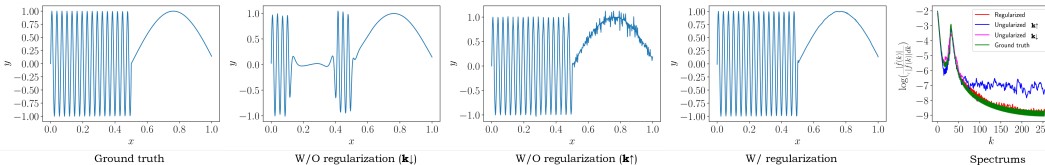

Figure 3: **The effect of explicit regularization.** When trained under conventional means, coordinate MLPs cannot generalize well at both higher and lower ends of the spectrum. This hinders the generalization performance of coordinate-MLPs when the target signal comprises regions with different spectral properties. When the network has insufficient bandwidth, the network cannot correctly capture high-frequency modes. When the network is tuned to have a higher bandwidth, the network fails at modeling lower frequencies. In contrast, the network can preserve a better spectrum with the proposed regularization scheme. This example uses a 4-layer sinusoid-MLP trained with 33% of the total samples.

## 7 Regularizing coordinate-MLPs

The spectrum of a function is inherently related to the magnitude of its derivatives. For instance, consider a function $f : \mathbb{R} \to \mathbb{R}$ defined on a finite interval $\epsilon$. Then, it can be shown that $\max_{x \in \epsilon} |\frac{df(x)}{dx}| \leq |2\pi| \int_{\infty}^{\infty} |k\hat{f}(k)| dk$ (see Appendix). For multi-dimensions, we can encourage the spectrum to have a higher or lower frequency support by appropriately constraining the fluctuations along the corresponding directions. We shall now discuss how this fact may be utilized in regularizing coordinate-MLPs.

Let us consider a coordinate-MLP $f : \mathbb{R}^n \to \mathbb{R}$, which we factorise as $f = \tilde{f} \circ g$, where $\tilde{f}$ is the final layer. Then by the chain rule:

$$\left| \frac{df}{d\mathbf{x}} \right| = \sqrt{\left[ \frac{\partial \tilde{f}}{\partial \mathbf{y}} \cdot \frac{\partial g}{\partial \mathbf{x}} \right] \left[ \frac{\partial \tilde{f}}{\partial \mathbf{y}} \cdot \frac{\partial g}{\partial \mathbf{x}} \right]^T} = \sqrt{\frac{\partial \tilde{f}}{\partial \mathbf{y}} \cdot \mathbf{J}\mathbf{J}^T \cdot \left[ \frac{\partial \tilde{f}}{\partial \mathbf{y}} \right]^T}, \tag{10}$$

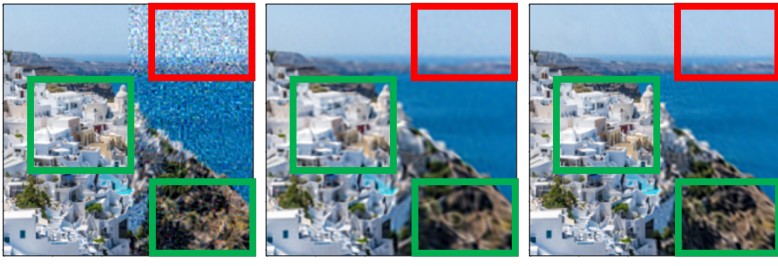

| W/O Regularization ( $\mathbf{k}\uparrow$ ) | W/O Regularization ( $\mathbf{k}\downarrow$ ) | W/ Regularization |

Figure 4: **Qualitative results for encoding signals with uneven sampling (zoom in for a better view)**. A Gaussian-MLP is trained to encode an image, where the left half of the image is sampled densely, and the right half is sampled with $10\%$ pixels. The reconstruction results are shown. When the bandwidth of the Gaussian-MLP is adjusted to match the sampling procedure of a particular half, the other half demonstrates poor reconstruction. In contrast, when regularized, coordinate-MLPs can preserve both low and high frequencies, giving a balanced reconstruction. Contrast the green and red areas across each setting.

where $\mathbf{J}$ is the Jacobian of $g$. Let $\mathbf{A} = \mathbf{J}\mathbf{J}^T$. In order to minimize $|\frac{df}{d\mathbf{x}}|$, it suffices to minimize the magnitude of the components of $\mathbf{J}$, which is equivalent to minimizing the trace of $\mathbf{A}$. Recall that $tr(A) = \sum_k \lambda_k$, where $\lambda_k$ are the eigenvalues of $\mathbf{A}$. One has $\lim_{|\epsilon| \to 0} \frac{|g(\mathbf{x}) - g(\mathbf{x} + \epsilon u_k)|}{|\epsilon u_k|} = \sqrt{\lambda_k}$, where $u_k$ are the eigenvectors of $\mathbf{A}$. Therefore, minimizing an eigenvalue of $\mathbf{A}$ is equivalent to restricting the fluctuations of $g$ along the direction of its associated eigenvector. By this logic, the ideal regularization procedure would be to identify the directions in the spectrum of the target signal where the frequency support is low, and restrict $\lambda_k$'s corresponding to those directions. According to our derivations in the previous sections, this would only be possible by regularizing $\mathbf{w}_i$'s, since in practice, we use hyperparameters for coordinate-MLPs that allow higher bandwidth. However, this is a cumbersome task and we empirically found that there is a much simpler approximation for this procedure that can give equivalent results (see Appendix) as,

$$\mathcal{L}_r = \frac{|g(\bar{\mathbf{x}}) - g(\bar{\mathbf{x}} + \xi)|}{|\xi|}, \tag{11}$$

where $\bar{\mathbf{x}}$ are randomly sampled from the coordinate space and $\xi \sim \mathcal{N}(0, \Sigma)$ where $\Sigma$ is diagonal with small values. The total loss function for the coordinate-MLP then becomes $\mathcal{L}_{total} = \mathcal{L}_{MSE} + \varepsilon \mathcal{L}_r$ where $\varepsilon$ is a small scalar coefficient and $\mathcal{L}_{MSE}$ is the ususal mean squared error loss. The total loss can also be interpreted as encouraging the networks to obtain the smoothest possible solution while perfectly fitting the training samples. By the same argument, one can apply the above regularization on an arbitrary layer, including the output, although we empirically observed that the penultimate layer performs best. Although the proposed loss has a similar goal to the popular *total-variation loss*, the former converges to more desired solutions (see Appendix).

## 8  Experiments

In this section, we will show that the insights developed thus far extend well to deep networks in practice.

### 8.1  Encoding signals with uneven sampling

The earlier sections showed that coordinate-MLPs tend to suppress low frequencies when the network attempts to add higher frequencies to the spectrum. This can lead to poor performance when the target signal is unevenly sampled because, if the sampling is dense, the network needs to incorporate higher frequencies to properly encode the signal. Thus, at regions where the sampling is sparse, the network fails to produce smooth interpolations, resulting in noisy reconstructions. On the other hand, if the bandwidth of the network is restricted via hyperparameters or depth, the network can smoothly interpolate in sparse regions, but fails to encode information with high fidelity at dense regions. Explicit regularization can aid the network in finding a properly balanced spectrum in the solution space. Fig. 3 shows an example for encoding a 1D signal. Fig. 4 illustrates a qualitative example in encoding a 2D image. Table 1 depicts quantitative results on the natural dataset by Tancik

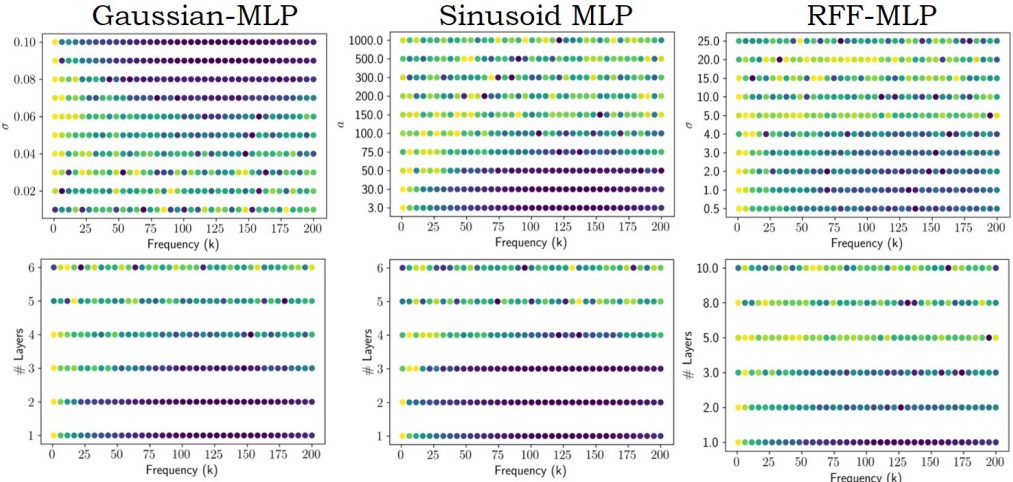

Figure 5: **Spectra of the coordinate-MLPs against hyperparameters and depth.** We train each network instance to encode a sound wave with 33% sampling. The heat indicates the intensity of the corresponding frequency component after trained with the MSE loss. As illustrated, when the capacity of the network is increased via hyperparameters or depth, the networks tend to converge to solutions with suppressed low-frequency components.

| W/O Regularization ($\mathbf{k} \downarrow$) | | | |
| --- | --- | --- | --- |
| Type | L-PSNR | R-PSNR | T-PSNR |
| Gaussian-MLP | 30.12 | 22.47 | 27.72 |
| Sinusoid-MLP | 30.49 | 21.93 | 26.89 |
| RFF-MLP | 29.89 | 22.33 | 26.44 |
| W/O Regularization ($\mathbf{k} \uparrow$) | | | |
| Type | L-PSNR | R-PSNR | T-PSNR |
| Gaussian-MLP | 33.43 | 18.14 | 22.80 |
| Sinusoid-MLP | 32.17 | 19.30 | 23.49 |
| RFF-MLP | 32.24 | 19.11 | 22.19 |
| Regularized | | | |
| Type | L-PSNR | R-PSNR | T-PSNR |
| Gaussian-MLP | 33.11 | 23.31 | 30.15 |
| Sinusoid-MLP | 31.59 | 22.66 | 29.94 |
| RFF-MLP | 31.99 | 22.91 | 29.59 |

Table 1: **Encoding images with uneven sampling.** In each training instance, the left half of the image is sampled densely, and the right half is sampled with 10% pixels. With unregularized coordinate-MLPs, when the hyperparameters are tuned to match the sampling of a particular half, the reconstruction of the other half is poor. In contrast, the encoding performance is balanced when regularized. We use 4-layer networks for this experiment.

et al. [2020]. L-PSNR, R-PSNR, and T-PSNR means PSNR evaluated on the left, right and the total image, respectively.

## 8.2 Encoding signals with different local spectral properties

The difficulty in generalizing well across different regions in the spectrum hinders encoding natural signals such as images when the sampling is sparse, as they tend to contain both "flat" and "fluctuating" regions, and the network has no prior on how to interpolate in these different regions. See Fig. 6 and Table 2 for qualitative and quantitative results. As evident, the coordinate-MLP struggles to smoothly interpolate between samples under a single hyperparameter setting. We also assess the performance of the regularization on NeRF style data. The results are depicted in Fig. 7 and Table. 3.

## 8.3 Effect of increasing capacity

Our derivations in Sec. 4 and 6 showed that shallow coordinate-MLPs tend to suppress lower frequencies when the capacity of the network is increased via depth or hyperparameters. Our empirical results show that this is indeed the case for deeper networks as well (see Fig. 5).

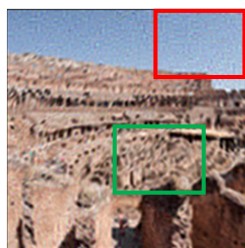 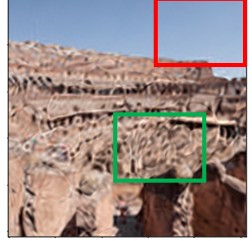 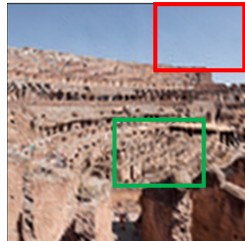

W/O Regularization (**k**↑)  W/O Regularization (**k**↓)  W/ Regularization

Figure 6: **A single hyperparameter setting cannot generalize well across the spectrum unless regularized (better viewed in zoom).** With sparse sampling (10% of the total pixels), coordinate-MLPs exhibit inferior reconstruction performance across regions with different spectra. In comparison, regularized networks can achieve the best of both worlds. Compare the highlighted areas across each setting.

| STL | | |
|---|---|---|
| Type | W/O Regularization | Regularized |
| Gaussian-MLP | 22.11 | 24.17 |
| Sinusoid-MLP | 21.97 | 25.01 |
| RFF-MLP | 22.03 | 23.31 |
| ImageNet | | |
| Gaussian-MLP | 24.59 | 26.77 |
| Sinusoid-MLP | 23.81 | 26.90 |
| RFF-MLP | 23.11 | 25.94 |

Table 2: **Encoding images with sparse sampling.** We compare the performance over the STL dataset Coates et al. [2011] and a sub-sampled version of ImageNet with 10% sampling. Regularized coordinate-MLPs show superior performance due to better interpolation properties. We use 4-layer coordinate-MLPs for this experiment.

| Loss | PSNR | SSIM |
|---|---|---|
| MSE + L1 | 30.91 | 0.947 |
| MSE + L2 | 30.14 | 0.935 |
| MSE | 30.87 | 0.940 |
| MSE+TV loss | 29.11 | 0.909 |
| MSE+proposed reg. | **32.01** | **0.951** |

Table 3: **Quantitative comparison in novel view synthesis on the real synthetic dataset [Mildenhall et al., 2020].** The proposed regularization achieves better results. In contrast, TV loss worsens the performance. Note that we use TV loss with 0.001 weight, and its performance converges to vanilla MSE loss as the weight gets smaller. L1 regularization improves the results slightly, but still is inferior to the proposed regularizer.

W/ Reg    W/O Reg

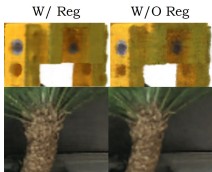

Figure 7: **Our regularization is able to achieve better reconstructions (NeRF data).** *Top row:* Proposed regularization suppresses the unnecessary higher frequencies. *Bottom row:* we can obtain sharper results by increasing the frequency support of the positional embedding layer and then applying the regularization. Cropped and zoomed in for better viewing.

## 9 Limitations

Currently, our work theoretically analyzes coordinate-networks within a constrained shallow setting. Although the insights we gather from this restricted setting can be empirically extrapolated to deeper networks (as we have shown), future research should focus on deriving rigorous formulae to analyze practical, deeper coordinate networks. Further, the current approach involves manually tuning the weights of the loss components, which requires a fair amount of domain knowledge. Perhaps, a more in-depth study can reveal general guidelines for tuning these parameters.

## 10 Conclusion

We show that the traditional implicit regularization assumptions do not hold in the context of coordinate-MLPs. We focus on establishing plausible reasoning for this phenomenon from a Fourier angle and discover that coordinate-MLPs tend to suppress lower frequencies when the capacity is increased unless explicitly regularized. We further show that the developed insights are valid in practice.

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
