# A  Appendix

## A.1  Proof for Theorem 4.1

In order to carry out the computation, we require the following lemma.

**Lemma A.1.** *Let $A : \mathbb{R}^{n-1} \to \mathbb{R}^n$ be an isometric linear map, and let $A^\perp \subset \mathbb{R}^n$ denote the 1-dimensional subspace which is orthogonal to the range of $A$. Then for any integrable function $\varphi \in L^1(\mathbb{R}^n)$, one has*

$$\int_{\mathbb{R}^{n-1}} \hat{\varphi}(A\mathbf{x}) d\mathbf{x} = (2\pi)^{\frac{n-1}{2}} \int_{A^\perp} \varphi(\mathbf{y}) \, d_{A^\perp} \mathbf{y},$$

*where $d_{A^\perp} \mathbf{y}$ is the volume element induced on the subspace $A^\perp$.*

*Proof.* First using the dominated convergence theorem, and then applying Fubini's theorem, we have

$$\int \hat{\varphi}(A\mathbf{x}) \, d\mathbf{x} = \lim_{\epsilon \to 0} \int e^{-\frac{\epsilon |\mathbf{x}|^2}{2}} \hat{\varphi}(A\mathbf{x}) \, d\mathbf{x}$$

$$= \lim_{\epsilon \to 0} \int \left( \int e^{-i(A\mathbf{x}) \cdot \mathbf{k}} e^{-\frac{\epsilon |\mathbf{x}|^2}{2}} \, d\mathbf{x} \right) \varphi(\mathbf{k}) \, d\mathbf{k}$$

$$= (2\pi)^{\frac{n-1}{2}} \lim_{\epsilon \to 0} \int \frac{e^{-\frac{|A^T \mathbf{k}|^2}{2\epsilon}}}{\epsilon^{\frac{n-1}{2}}} \varphi(\mathbf{k}) \, d\mathbf{k},$$

where on the last line we have used the fact that the Fourier transform of a Gaussian is again a Gaussian. Now, for any $\mathbf{y} \in A^\perp$, let $A^\parallel(\mathbf{y})$ denote the $(n-1)$-dimensional hyperplane parallel to the range of $A$ that passes through $\mathbf{y}$, and denote by $d_{A^\parallel(\mathbf{y})} \mathbf{z}$ the induced volume element along $A^\parallel(\mathbf{y})$. Each $\mathbf{k} \in \mathbb{R}^n$ decomposes uniquely as $\mathbf{k} = \mathbf{y} + \mathbf{z}$ for some $\mathbf{y} \in A^\perp$ and $\mathbf{z} \in A^\parallel(\mathbf{y})$. The integral over $\mathbb{R}^n$ of any integrable function $\psi$ on $\mathbb{R}^n$ can then be written

$$\int \psi(\mathbf{k}) \, d\mathbf{k} = \int_{A^\perp} \int_{A^\parallel(\mathbf{y})} \psi(\mathbf{y} + \mathbf{z}) d_{A^\parallel(\mathbf{y})} \mathbf{z} \, d_{A^\perp} \mathbf{y}$$

Since the function $e^{-\frac{|A^T \mathbf{k}|^2}{2\epsilon}}$ is constant along any line parallel to $A^\perp$, the dominated convergence theorem now tells us that $(2\pi)^{\frac{1-n}{2}} \int \hat{\varphi}(A\mathbf{x}) \, d\mathbf{x}$ is equal to

$$\int_{A^\perp} \lim_{\epsilon \to 0} \int_{A^\parallel(\mathbf{y})} \frac{e^{-\frac{|A^T(\mathbf{z})|^2}{2\epsilon}}}{\epsilon^{\frac{n-1}{2}}} \varphi(\mathbf{y} + \mathbf{z}) \, d_{A^\parallel(\mathbf{y})} \mathbf{z} \, d_{A^\perp} \mathbf{y}$$

$$= \int_{A^\perp} \lim_{\epsilon \to 0} \int_{\mathbb{R}^{n-1}} \frac{e^{-\frac{|\mathbf{x}|^2}{2\epsilon}}}{\epsilon^{\frac{n-1}{2}}} \varphi(\mathbf{y} + A\mathbf{x}) \, d\mathbf{x} \, d_{A^\perp} \mathbf{y}$$

$$= \int_{A^\perp} \lim_{\epsilon \to 0} (\eta_\epsilon * \varphi_{\mathbf{y}})(0) \, d_{A^\perp} \mathbf{y}$$

$$= \int_{A^\perp} \varphi(\mathbf{y}) d_{A^\perp} \mathbf{y},$$

where on the third line we have used the definitions $\varphi_{\mathbf{y}}(\mathbf{x}) := \varphi(\mathbf{y} + A\mathbf{x})$ and $\eta_\epsilon(\mathbf{x}) := \epsilon^{-\frac{n-1}{2}} e^{-\frac{|\mathbf{x}|^2}{2\epsilon}}$ of functions on $\mathbb{R}^{n-1}$, and on the final line we have used the fact that convolution with $\eta_\epsilon$ is an approximate identity for $L^1(\mathbb{R}^{n-1})$. $\square$

**Theorem A.2.** *The Fourier transform of $f$ is the distribution*

$$\hat{f}(\mathbf{k}) = \frac{(2\pi)^{\frac{n}{2}}}{|\mathbf{w}|} \hat{\alpha}\left( \frac{\mathbf{w}}{|\mathbf{w}|^2} \cdot \mathbf{k} \right) \delta_{\mathbf{w}}(\mathbf{k}),$$

*where $\delta_{\mathbf{w}}(\mathbf{k})$ is the Dirac delta distribution which concentrates along the line spanned by $\mathbf{w}$.*

*Proof.* Let $A = \left[\frac{\mathbf{w}}{|\mathbf{w}|}, \mathbf{a_2}, \ldots, \mathbf{a_n}\right]$ be any special orthogonal matrix for which $A\mathbf{e_1} = \frac{\mathbf{w}}{|\mathbf{w}|}$, letting $\bar{A} = [\mathbf{a_2}, \ldots, \mathbf{a_n}]$ denote the corresponding submatrix. Given a vector $\mathbf{k} = (k_1, \ldots, k_n)^T \in \mathbb{R}^n$, use the notation $\mathbf{k}' = (k_2, \ldots, k_n)^T \in \mathbb{R}^{n-1}$. Then the pairing $\langle f, \hat{\varphi} \rangle$ is given by

$$(2\pi)^{-\frac{n}{2}} \int \int e^{-i\mathbf{x}\cdot\mathbf{k}} \alpha(\mathbf{w}\cdot\mathbf{x})\varphi(\mathbf{k}) \, d\mathbf{x} \, d\mathbf{k}$$

$$=(2\pi)^{-\frac{n}{2}} \int \int e^{-i\mathbf{y}\cdot(A^T\mathbf{k})} \alpha(\mathbf{w}\cdot(A\mathbf{y}))\varphi(\mathbf{k}) \, d\mathbf{y} \, d\mathbf{k}$$

$$=(2\pi)^{\frac{1-n}{2}} \int \int \left( (2\pi)^{-\frac{1}{2}} \int e^{-iy_1 \frac{\mathbf{w}}{|\mathbf{w}|}\cdot\mathbf{k}} \alpha(|\mathbf{w}|y_1) \, dy_1 \right)$$
$$\times e^{-i\mathbf{y}'\cdot(A^T\mathbf{k})'} \varphi(\mathbf{k}) \, d\mathbf{y}' \, d\mathbf{k}$$

$$=(2\pi)^{\frac{1-n}{2}} \int \int \frac{1}{|\mathbf{w}|}\hat{\alpha}\left(\frac{\mathbf{w}}{|\mathbf{w}|^2}\cdot\mathbf{k}\right)\varphi(\mathbf{k})e^{-i\mathbf{y}'\cdot(A^T\mathbf{k})'} \, d\mathbf{k} \, d\mathbf{y}'$$

$$=(2\pi)^{\frac{1-n}{2}} \int \int \frac{1}{|\mathbf{w}|}\hat{\alpha}\left(\frac{\mathbf{w}}{|\mathbf{w}|^2}\cdot\mathbf{k}\right)\varphi(\mathbf{k})e^{-i\mathbf{y}'\cdot(\bar{A}^T\mathbf{k})} \, d\mathbf{k} \, d\mathbf{y}'$$

$$=(2\pi)^{\frac{1-n}{2}} \int \int \frac{1}{|\mathbf{w}|}\hat{\alpha}\left(\frac{\mathbf{w}}{|\mathbf{w}|^2}\cdot\mathbf{k}\right)\varphi(\mathbf{k})e^{-i(\bar{A}\mathbf{y}')\cdot\mathbf{k}} \, d\mathbf{k} \, d\mathbf{y}'$$

$$=(2\pi)^{\frac{1}{2}} \int \hat{g}(\bar{A}\mathbf{y}') \, d\mathbf{y}',$$

where for the first equality we have made the substitution $\mathbf{x} = A\mathbf{y}$, and in the final step we have set $g(\mathbf{k}) := \frac{1}{|\mathbf{w}|}\hat{\alpha}\left(\frac{\mathbf{w}}{|\mathbf{w}|^2}\cdot\mathbf{k}\right)\varphi(\mathbf{k})$. Our integral is thus of the form given in Lemma A.1, and the result follows. $\square$

## A.2  First order gradient magnitudes vs. Fourier spectrum

Consider a signal $f$. Then,

$$f(x) = \int_{\infty}^{\infty} \hat{f}(k)e^{2\pi ikx} dk$$

It follows that,

$$|\frac{df(x)}{dx}| = |2\pi i \int_{\infty}^{\infty} k\hat{f}(k)e^{2\pi ikx} dk| \tag{12}$$

$$\leq |2\pi| \int_{\infty}^{\infty} |k\hat{f}(k)| dk. \tag{13}$$

Therefore,

$$\max_{x\in\epsilon} |\frac{df(x)}{dx}| \leq |2\pi| \int_{\infty}^{\infty} |k\hat{f}(k)| dk. \tag{14}$$

## A.3  Fourier transform of composite functions

Let a stack of layers be denoted as $\kappa : \mathbb{R}^n \to \mathbb{R}^d$. Suppose we add another stack of layers $\eta : \mathbb{R}^d \to \mathbb{R}$ on top of $\kappa(\cdot)$ to construct the MLP $\eta \circ \kappa : \mathbb{R}^n \to \mathbb{R}$. One can rewrite $(\eta \circ \kappa)$ in terms of the inverse Fourier transform as,

$$(\eta \circ \kappa) = (\frac{1}{\sqrt{2\pi}})^d \int_{\mathbb{R}^d} \eta(\mathbf{t})e^{i\mathbf{t}\cdot\kappa(\mathbf{x})} d\mathbf{t}$$

Fourier transform of $(\eta \circ \kappa)$ now can be written as,

$$\widehat{(\eta \circ \kappa)} = \left(\frac{1}{\sqrt{2\pi}}\right)^{d+n} \int_{\mathbb{R}^n} \int_{\mathbb{R}^d} \eta(\mathbf{t}) e^{i\mathbf{t}\cdot\kappa(\mathbf{x})} d\mathbf{t} e^{-i\mathbf{k}\cdot\mathbf{x}} d\mathbf{x}$$

$$= \left(\frac{1}{\sqrt{2\pi}}\right)^{d+n} \int_{\mathbb{R}^d} \eta(\mathbf{t}) \int_{\mathbb{R}^n} e^{i\mathbf{l}\cdot\kappa(\mathbf{x})} e^{-\mathbf{k}\cdot\mathbf{x}} d\mathbf{x} d\mathbf{t},$$

which yields,

$$\widehat{(\eta \circ \kappa)}(\mathbf{k}) = \left(\frac{1}{\sqrt{2\pi}}\right)^n \langle \hat{\eta}(\cdot), \beta(\mathbf{k}, \cdot)\rangle,$$

where, $\beta(\mathbf{k}, \mathbf{t}) = \int_{\mathbb{R}^n} e^{i(\mathbf{t}\cdot\kappa(\mathbf{x})-\mathbf{k}\cdot\mathbf{x})}$. Bergner et al. [2006] showed that in the Fourier transforms of above form, the maximum composite frequency is,

$$\mathbf{k}^*_{\eta\circ\kappa} = \max_{|\mathbf{u}|=1}(\mathbf{k}^*_{\hat{\eta},\mathbf{u}} \max_{\mathbf{x}}(\mathbf{u} \cdot \kappa'(\mathbf{x}))).$$

Considering the 1D case, at the stationary phase of $\beta$, it can be seen that $\frac{du}{dx} = 0$, where $u(x) = t\kappa(x) - kx = 0$. Converting to polar coordinates,

$$\frac{du}{dx} = \frac{d}{dx}r(\kappa(x)\sin\theta - x\cos\theta) = 0$$

$$\kappa'(x_s)\sin\theta - \cos\theta = 0$$

$$\frac{1}{\kappa'(x_s)} = \tan\theta,$$

where $x_s$ are the points at stationary phase. By using the Taylor approximation around $x_s$, we get,

$$I_{x_s} \sim \int_{\infty}^{\infty} e^{r(\kappa(x_s)\sin\theta - x_s\cos\theta + \frac{1}{2}\kappa''(x_s)x^2\sin\theta)} dx$$

$$I_{x_s} \sim \int_{\infty}^{\infty} e^{r(\kappa(x_s)\sin\theta - x_s\cos\theta)} \left(\frac{2\pi}{r|\kappa''(x_s)\sin\theta|}\right)^{\frac{1}{2}} e^{i\frac{\pi}{4}sgn\{\kappa''(x_s)\sin\theta\}}$$

At points where $\kappa''(x_s)$ differs from 0 largely, the integral vanishes as $(x - x_s)^2$ increases. Since we can obtain the full integral by summing $I_{x_s}$ for all $x_s$ such that $\frac{1}{\kappa'(x_s)} = \tan\theta$. Hence, around the stationary phase,

$$\min(\kappa') < \frac{1}{\tan\theta} < \max(\kappa')$$

which is essentially,

$$\min(\kappa') < \frac{k}{l} < \max(\kappa').$$

## A.4  Multidimensional RFF

Let $\Omega$ denote the probability space $\mathbb{R}^n$ equipped with the Gaussian measure of standard deviation $2\pi\sigma > 0$. For $D \geq 1$, write elements $\mathbf{L}$ of $\Omega^n$ as matrices $[\mathbf{l}_1, \ldots, \mathbf{l}_D]$ composed of $D$ vectors drawn independently from $\Omega$. Consider the random Fourier feature (RFF) positional embedding $f_1 : \Omega^D \times \mathbb{R}^n \to \mathbb{R}^{2D}$

$$f_1(\mathbf{L}, \mathbf{x}) := [\mathbf{e}(\mathbf{l}_1, \mathbf{x}), \ldots, \mathbf{e}(\mathbf{l}_D, \mathbf{x})]^T,$$

where $\mathbf{e} : \Omega \times \mathbb{R}^n \to \mathbb{R}^2$ is the random function defined by the formula

$$\mathbf{e}(\mathbf{l}, \mathbf{x}) := [\sin(\mathbf{l} \cdot \mathbf{x}), \cos(\mathbf{l} \cdot \mathbf{x})].$$

The positional embedding is followed by a linear transformation $A : \mathbb{R}^{2D} \to \mathbb{R}^{2D}$ yielding

$$f_2(\mathbf{L}, \mathbf{x})^i = \sum_j a_j^i f_1(\mathbf{L}, \mathbf{x})^j.$$

We approximate the following ReLU by a Newman polynomial

$$N_m(x) := \prod_{i=1}^{m-1} \left(x + \exp\!\left(-i/\sqrt{n}\right)\right),$$

for $m \geq 5$, yielding

$$f_3(\mathbf{L}, \mathbf{x})^i = \prod_{\alpha=1}^{m-1} \left( \sum_j a_j^i f_1(\mathbf{L}, \mathbf{x})^j + \exp\!\left(-\alpha/\sqrt{n}\right) \right)$$

$$= \sum_{\beta=0}^{m-2} \kappa(m, \beta) \Big( \sum_j a_j^i f_1(\mathbf{L}, \mathbf{x})^j \Big)^\beta$$

$$= \sum_{\beta=0}^{m-2} \kappa(m, \beta) \times$$

$$\sum_{k_1 + \cdots + k_{2D} = \beta} \binom{\beta}{k_1, \ldots, k_{2D}} \prod_{t=1}^{2D} (a_t^i f_1(\mathbf{L}, \mathbf{x})^t)^{k_t}$$

where $\kappa(m, \beta) = \binom{m-2}{\beta} \prod_{\alpha=1}^{m-1-\beta} \exp(-\alpha/\sqrt{m})$. It follows that the spectrum of $f_3$ is determined by the spectra of the powers $\left(f_1(\mathbf{L}, \mathbf{x})^j\right)^k$ of the components of $f_1$. Write $j = 2j_1 + j_2$, where $j_1$ is the floor of $j/2$. Then

$$\gamma(\mathbf{L}, \mathbf{x})^j = \frac{e^{i \mathbf{l}_{j_1} \cdot \mathbf{x}} + (-1)^j e^{i \mathbf{l}_{j_1} \cdot \mathbf{x}}}{2}.$$

Therefore, for $k \geq 0$,

$$(\gamma(\mathbf{L}, \mathbf{x})^j)^k = 2^{-k} \sum_{a=0}^{k} (-1)^{j(k-a)} e^{i(2a-k)\mathbf{l}_{j_1} \cdot \mathbf{x}}.$$

We then see that

$$\prod_{t=1}^{2D} (f1(\mathbf{l}, \mathbf{x})^t)^{k_t} = \prod_{t=1}^{2D} 2^{-k_t} \left( \sum_{a=0}^{k_t} (-1)^{t(k-a)} e^{i(2a-k)\mathbf{l}_{j_1} \cdot \mathbf{x}} \right)$$

$$= C \sum_{a_1, \ldots, a_{2D}=0}^{k_1, \ldots, k_{2D}} \prod_{t=1}^{2D} (-1)^{t(k_t - a_t)} e^{i(2a_t - k_t)\mathbf{l}_{t_1} \cdot \mathbf{x}}$$

$$= C \sum_{a_1, \ldots, a_{2D}=0}^{k_1, \ldots, k_{2D}} C' e^{i \sum_{t=1}^{2D} (2a_t - k_t)\mathbf{l}_{t_1} \cdot \mathbf{x}},$$

where $C = \prod_{t=1}^{2D} 2^{-k_t}$ and $C' = \prod_{t=1}^{2D} (-1)^{t(k_t - a_t)}$. The proof of Theorem A.2 can then be used to show that the Fourier transform of $f_3$ is a linear combination of terms of the form

$$\mathbf{k} \mapsto \delta\!\left( \frac{\mathbf{s} \cdot \mathbf{k}}{|\mathbf{s}|^2} - 1 \right) \delta_{\mathbf{s}}(\mathbf{k}),$$

where $\mathbf{s} = \sum_{j=1}^{D} \left( (2a_{2j-1} - k_{2j-1}) + (2a_{2j} - k_{2j}) \right) \mathbf{l}_j$ for positive integers $k_1, \ldots, k_{2D}$ such that $k_1 + \cdots + k_{2D} \leq m - 2$, and integers $0 \leq a_j \leq k_j$ for $j = 1, \ldots, 2D$. It follows that the spectrum of $f_3$ is concentrated on frequencies

$$\mathbf{k} = \sum_{j=1}^{D} \left( (2a_{2j-1} - k_{2j-1}) + (2a_{2j} - k_{2j}) \right) \mathbf{l}_j,$$

and that therefore the spectrum can be made wider, with high probability, by increasing the standard deviation $\sigma$ of the distribution from which the $\mathbf{l}_j$ are drawn.

## A.5    F-principle

F-principle is a phenomenon which states that DNNs tend to learn lower frequencies first. One of the earliest works on this was presented by Xu et al. [2019] and Rahaman et al. [2019]. Follow up theoretical studies indicate that the F-Principle holds under general setting with infinite samples [Luo et al., 2019] and in the NTK regime with finite samples [Zhang et al., 2019, Luo et al., 2020] or samples distributed uniformly on sphere [Cao et al., 2019, Bordelon et al., 2020]. We observed that F-principle is also valid for coordinate-MLPs. Fig. 8 shows an empirical example.



| Epoch 100 | Epoch 500 | Epoch 1000 | Epoch 2000 | Epoch 5000 |

Figure 8: **The F-principle holds for coordinate-MLPs.** Above frequency plots show the difference between the network's spectrum and the target spectrum while training. As evident, the network learns the low frequencies first.

## A.6    Total variation (TV) loss vs the proposed regularization

At a high-level, the goals of TV-loss and the proposed regularization are similar. However, by regularising directly on the network outputs, TV regularisation necessarily introduces a trade-off between perfectly fitting the training data (i.e. ensuring network outputs given data are correct) and reducing variation between adjacent data points. In contrast, our method *1)* explicitly regularises based *only* on random samples taken *between* data points, and *2)* depends only on the outputs of the penultimate layer. Since the final layer is linear, our regularisation therefore affords greater capacity to fit training data while simultaneously encouraging smooth (linear) interpolation between training points, lessening the tradeoff between these objectives present in TV. Our regularization shows superior performance over TV both qualitatively and quantitatively.

## A.7    Comparison with SDF regularisation in Sitzmann et al. [2020]

In their work, Sitzmann et al. [2020] proposed a regularization scheme tailored for 3D signed distance function (SDF) problems. In particular, they enforced that the gradient magnitude of the output with respect to the coordinate is regularised should be 1. One can be misled that our regularization contradicts the above, since ours try to minimize the gradients overall. However, it is straightforward to show that our regularisation induces the same solution Sitzmann et al. [2020], while being applicable to a larger set of problem domains. Consider a line drawn along a normal vector to the surface of interest in 3D space. Training requires that the network outputs, representing the SDF, increase linearly (in magnitude) on training points along this line. Since our method regularises only on the penultimate layer, *between* the training points, and the final layer is linear, our regularisation encourages a *linear interpolant* between these training data. Because there is precisely one linear interpolant which fits training data along this line, our regularisation will encourage convergence to the same solution as the regularisation proposed by Sitzmann et al. [2020].

| Regularization | Avg. training time (S) | Performance |
|---|---|---|
| Eigen directions | 1194.96 | 24.77 |
| Random directions | 127.56 | 24.17 |

Table 4: We compare the proposed approximate regularizer vs. the idea regularizer (that minimizes the fluctuations along eigen vectors). We used the STL dataset with 10% training sampling. As shown, we the ideal regularizer takes significantly more training time while achieving marginal performance improvement. We use 4-layer Gaussian-MLPs for this experiment.