# OpenReview forum: "On the Frequency-bias of Coordinate-MLPs"
_NeurIPS.cc/2022/Conference — NeurIPS 2022 Accept_

### Official Review · Reviewer_4Raw · 2022-06-21

**Rating:** 5
**Confidence:** 4
**Soundness:** 2 fair
**Presentation:** 4 excellent
**Contribution:** 2 fair

**Summary:**

One-liner: Authors study the spectral bias of INRs, and propose a regularizer to fix it.

This paper argues that popular neural network architectures for INRs---or coordinate-MLPs, as authors call it---tend to learn functions with "suppressed" low frequency components. To support this claim, the paper provides (i) Fourier transforms of the function expressed by (the two-layer versions of) these architectures which hints the effect of depth and hyperparameters on low-and-high frequency components, and (ii) empirical results showing that the spectra of the functions learned by these architectures depend on the hyperparameters and depth. Based on this observation, authors propose a (randomized) regularizer which aims to amplify the low-frequency component of the learned signal.

**Questions:**

Here is an additional question, that I do not necessarily view as a weakness but very important to answer nevertheless.

- One of the critical architectural factor has not been discussed---width! From my understanding, width is one of the most critical factors about the implicit regularizations of overparameterized neural networks (not confined to INR context). In fact, I believe that width is often treated as a more important factor for generalization than depth. Indeed, I did observe from my own experimental results that widening the networks tend to have beneficial impact on the super-resolution capabilities of INRs. I wonder if the authors' analytical tools can be useful in studying the effects of width. This should help strengthening the main message of the paper.

**Limitations:**

I couldn't find any explicit discussions on limitations or potential negative impact, but I do not see this as a big drawback, because I also could not see a very clear potential negative societal impact of this work.

**Strengths And Weaknesses:**

- A strength of this paper is its section 7, where the authors design a regularizer motivated by the spectral understanding of the network. Although I do not agree to some points (e.g., the tightness of the lower bound, as being sharp is different from being tight), I do find the approach original and interesting.

- Another thing that I like about this paper is the quality of writing. I appreciate the authors' effort on distinguishing what can be shown rigorously and what cannot be, e.g., as in Line 255.

- Also, the message conveyed in the paper is potentially very meaningful: "Conventional way of boosting the expressivity of INRs may be at odds with boosting the generalization performance, unlike as in conventional model architectures."

- A weak point is that some of the main claims are not properly formalized or ill-supported. As an example, authors argue that 'vanilla INRs are incapable of simultaneously generalizing well at both lower and higher ends of the spectrum,' or 'INRs do not naturally converge to solutions with rich spectrums in both low and high frequencies.' Here, I am not really sure what authors mean---I do not understand the ideal property the authors are looking for. From figure 1 (and later figures), I think that authors are actually looking for architectural bias that 'prefers low-frequency solutions whenever a perfect fit is possible,' instead of one that is frequency-neutral. This might be the reason why I was a bit puzzled when I saw Figure 6, where the authors claim that INRs have "suppressed" low freq. From the figure, I cannot really tell---the color distribution seems to be quite uniform on SIREN and RFF, given deep enough depth and large enough hyperparameters? I cannot imagine the "ideal heatmap" the authors have in mind, so that the SIREN/RFF has a relatively smaller low freq components. This part should be clear.

- In my opinion, theoretical contribution of this paper is rather weak. The paper does a nice job in presenting two-layer Fourier transforms of INRs, but I do not think they directly imply the main message. Especially, the arguments in section 6 (depth) is quite hand-wavy and not persuasive. In this aspect, I suggest that authors put more emphasis on describing the empirical observations and contributions.

- As theoretical analysis does not give any decisive conclusion, I think that the the experimental section needs to be more emphasized. Section 8 lacks a proper baseline---one should test the effectiveness of the proposed regularizer by comparing against existing regularizers, e.g., weight decay. In fact, it seems very likely that simply performing the weight decay may help boosting the low-frequency component of the learned signal, as model weights are what is being used as "frequency," especially in SIRENs.

- A nitpick is that it is very difficult to read the text on most figures if I print the paper.

- Nitpick#2: Line 75: "this effect is common to all the types of coordinate-MLPs." This cannot be true, as there are much more different architectures for INRs, e.g., multiplicative filter networks.

- Nitpick#3: I do not agree to the authors' point that previous mainstream works focused more on the contribution of SGD on the model bias than the architectural bias. In fact, while the term "implicit bias" or "implicit regularization" is being almost exclusively used for describing the work about how GD/SGD shapes the learned function, there are still many works on the impact of model architectures. For instance, there are plenty of works regarding ConvNet bias vs. Transformer bias, e.g., [ConViT](https://arxiv.org/abs/2103.10697). I do not even think that Zhang et al.'s main message was that implicit bias is "not" tied to the architecture.

---

> ### Author Response · Authors · 2022-08-02
> **Response to Reviewer 4Raw Part 1**
>
> We appreciate the thoughtful comments and insights. Please find our answers below.
>
> **1) A strength of this paper is its section 7, where the authors design a regularizer motivated by the spectral understanding of the network. Although I do not agree to some points (e.g., the tightness of the lower bound, as being sharp is different from being tight), I do find the approach original and interesting.**
>
> The reviewer is correct. We agree that statement regarding the tightness of the bound might not be entirely accurate. We have fixed this in revision.
>
> **2) A weak point is that some of the main claims are not properly formalized or ill-supported. As an example, authors argue that 'vanilla INRs are incapable of simultaneously generalizing well at both lower and higher ends of the spectrum,' or 'INRs do not naturally converge to solutions with rich spectrums in both low and high frequencies.' Here, I am not really sure what authors mean**
>
> The statement “INRs do not naturally converge to solutions with rich spectrums in both low and high frequencies” means that if the training labels have both higher and lower fluctuations, given enough capacity, INRs tend to capture the higher fluctuations, but as a result, they fail to smoothly interpolate between samples with lower fluctuations. We have made this clearer in revision.
>
>
> **3) From the figure, I cannot really tell---the color distribution seems to be quite uniform on SIREN and RFF, given deep enough depth and large enough hyperparameters? I cannot imagine the "ideal heatmap" the authors have in mind, so that the SIREN/RFF has a relatively smaller low freq components. This part should be clear.**
>
> One should interpret Figure 6 by observing the trend of suppression of the lower frequencies when the number of layers is increased, or parameters are changed to increase the frequency capacity. This trend is visible in all the architectures. We agree that the reviewer’s confusion is fair and we should have more clearly explained this. We will add ground truth frequency plots to compare against in the final version so that the message will be clearer.
>
> **4)  In my opinion, theoretical contribution of this paper is rather weak.**
>
> We agree that the theoretical results only consider two-layer networks. However, please note that rigorous characterization of deep networks is a difficult problem that involves extremely high-dimensional parameter spaces which can be intractable. Therefore, it is a common practice in the machine learning literature (especially when studying generalization properties of NNs) to study neural networks under restrictive assumptions. Few such examples include the study of linear networks [1], shallow networks [2], wide networks [3], vanishing initializations [4], or infinitesimal learning rates [5]. We also adopted a similar approach and later empirically extended the insights to a practical setting.
>
>
> [1] - Soudry et al. "The implicit
> bias of gradient descent on separable data." JMLR, 2018.
>
> [2] - Gunasekar et al. "Characterizing implicit bias in terms
> of optimization geometry." ICML, 2018
>
> [3] -  Mei et al. "Mean-field theory of two-layers neural
> networks: dimension-free bounds and kernel limit." ICLR, 2019
>
> [4] -  Chizat et al., "On lazy training in differentiable programming." NeurIPS, 2019.
>
> [5] - Moroshko et al. "Implicit bias in deep linear classification: Initialization scale vs training accuracy." NeurIPS, 2020.
>
>
> **5) One should test the effectiveness of the proposed regularizer by comparing against existing regularizers, e.g., weight decay. In fact, it seems very likely that simply performing the weight decay may help boosting the low-frequency component of the learned signal, as model weights are what is being used as "frequency," especially in SIRENs.**
>
> This is an excellent point. We have already included a comparison with the TV-loss in Table 2, since it is a similar regularization technique to ours. Following the reviewer’s suggestion, we have now included a comparison with the weight decay in Table 3. As shown, we got slightly better results compared to MSE, but is still inferior to ours. We will include a more comprehensive ablation study in the final version.
>
> **6) Nitpick2: Line 75: "this effect is common to all the types of coordinate-MLPs." This cannot be true, as there are much more different architectures for INRs, e.g., multiplicative filter networks.**
>
> We agree with the reviewer.  We have modified this sentence in the revised version.

---

> > ### Comment · Reviewer_4Raw · 2022-08-07
> > **Thank you for the response**
> >
> > Thank you for the detailed response. I have raised a score (4 -> 5), mainly due to the promised empirical comparison with existing regularizers.

---

> ### Author Response · Authors · 2022-08-02
> **Response to Reviewer 4Raw Part 2**
>
> **7) Nitpick#3: I do not agree to the authors' point that previous mainstream works focused more on the contribution of SGD on the model bias than the architectural bias. In fact, while the term "implicit bias" or "implicit regularization" is being almost exclusively used for describing the work about how GD/SGD shapes the learned function, there are still many works on the impact of model architectures. For instance, there are plenty of works regarding ConvNet bias vs. Transformer bias, e.g., ConViT.**
>
> We are not in anyway disputing that there has been works that study the impact of architecture in generalization. However, according to our observation, a significant bulk of the theoretical works focus on SGD. To address the reviewers concern, we have changed the wording slightly in the revised version.
>
> **8) I do not even think that Zhang et al.'s main message was that implicit bias is "not" tied to the architecture.**
>
> We agree that the interpretation of Zhang et al.'s main message can be debatable. However,  we believe that it indeed promotes the effect of SGD over architecture in generalization. For instance, they analyze how SGD acts as an implicit regularizer for linear models. Specifically, they show that SGD always converges linear models to a solution with a small norm, and thus, the optimization algorithm itself implicitly regularizes the solution. Then they use a variety of deep architectures and show that each of these networks implicitly regularize, (in our opinion), implying that implicit generalization might be an architectural agnostic property. Neither they nor we, however, dispute the fact that some architectures might generalize better than others.
>
>
> **9) One of the critical architectural factor has not been discussed---width! From my understanding, width is one of the most critical factors about the implicit regularizations of overparameterized neural networks (not confined to INR context). In fact, I believe that width is often treated as a more important factor for generalization than depth. Indeed, I did observe from my own experimental results that widening the networks tend to have beneficial impact on the super-resolution capabilities of INRs. I wonder if the authors' analytical tools can be useful in studying the effects of width.**
>
> The reviewer is correct in assessing that the width plays an important part in the performance of an INR. Although we are currently unable to connect the width to generalization, we would like to offer an explanation --- not using the tools developed in our paper --- that would relate the width to an equally important aspect: memorization. Since memorization is an important aspect of superresolution, we hope that the following explanation will shed some light on the reviewer's observations.
>
> Consider an INR employed on $N$ points ${\textbf{x}_1, \textbf{x}_2, \dots, \textbf{x}_N}$ where $\textbf{x}_i \in \mathbb{R}^k$. Consider the function $\phi(\cdot):\mathbb{R}^k \to \mathbb{R}^d$ that maps the inputs to the output of the penultimate layer of an INR of width $D$. Now, consider the matrix,
>
> \begin{equation}
>     \textbf{X} \in \mathbb{R}^{D \times N}  = \begin{bmatrix}
>        \phi(\textbf{x}_1)^T \phi(\textbf{x}_2)^T \dots  \phi(\textbf{x}_N)^T\\
>     \end{bmatrix}
> \end{equation}
>
> Recall that the final layer of an MLP is (typically) an affine projection (with weight matrix $\textbf{A}$) without any non-linearity. Dropping the bias for simplified notation, we get,
>
> \begin{equation}
>     \tilde{\textbf{Y}} = \textbf{A}^T\textbf{X} ,
> \end{equation}
> where $\tilde{\textbf{Y}} \in \mathbb{R}^{q \times N}$ are the outputs of the MLP. Suppose $\textbf{Y} \in \mathbb{R}^{q \times N}$ are the ground truth training outputs the MLP is attempting to learn. Observe that if the MLP is perfectly memorizing the training set --- if $\tilde{\textbf{Y}} = \textbf{Y}$ --- then
>  each row of $\textbf{Y}$ is a linear combination of the rows of $\textbf{X}$. Assume we have no prior knowledge of $\textbf{Y}$, that is, the rows of $\textbf{Y}$ can be *any* arbitrary vector in $\mathbb{R}^N$. If the rows of $\textbf{X}$ are linearly independent, they form a basis for $ \mathbb{R}^N$ (assuming $D \geq N)$. Therefore, if $\mathrm{rank}(\textbf{X}) = N$, it is guaranteed that (assuming perfect convergence) the MLP  can find a weight matrix $\textbf{A}$ that ensures perfect reconstruction of $\textbf{Y}$.

---

> ### Author Response · Authors · 2022-08-02
> **Response to Reviewer 4Raw Part 3**
>
> Then, how can we effectively use INR to memorize signals  where $D \ll N$ in practice? Note that although the condition $\mathrm{rank}(\textbf{X}) = N$ is sufficient to ensure perfect memorization for *any* signal, it might not always be necessary since natural signals are typically redundant -- that is of limited bandwidth. The bandwidth of a category of signals can be defined as the number of linearly independent  (normalized) bases required to represent them. Thus, $\mathrm{rank}(\textbf{X})$ can be less than $N$ for many categories of natural signals (including natural images) whilst still enjoying perfect signal recovery by the MLP. In contrast,  encoding noise signals which have limited to no redundancy -- would require a larger network width -- and yields poorer results with $D \ll N$. Nevertheless, even with natural signals that contains redundancies, having a larger width $D$ would help in perfect memorization of training points since it improves the lower bound on tha rank($\textbf{X}$). This is important in superresolution applications, which aligns with the reviewer's observations.

---

### Official Review · Reviewer_F1QE · 2022-07-04

**Rating:** 6
**Confidence:** 3
**Soundness:** 3 good
**Presentation:** 2 fair
**Contribution:** 3 good

**Summary:**

The paper studies implicit regularization in implicit neural representation approaches, primarily coordinate-MLPs. The authors analyze failure modes of coordinate-MLPs by carrying out an analysis of shallow MLP models, investigating the effect of depth and hyperparameters on their spectral bias. A smoothness regularization method is introduced and evaluated on image encoding.

**Questions:**

* Coordinate-MLPs are finding application even outside of implicit representation of data. It is common to provide "coordinates" as additional inputs to neural network architectures, or to use periodic activation functions (see for example current architectures for diffusion models (1), or neural operators (2, 3) ). I am curious if the authors think the proposed regularized could be helpful even in general deep learning tasks.

(1) Ho et al., Denoising Diffusion Probabilistic Models

(2) Li et al., Fourier Neural Operator for Parametric Partial Differential Equations

(3) Poli et al., Transform Once: Efficient Operator Learning in Frequency Domain

**Limitations:**

The authors provided a satisfactory discussion on limitations and comparisons with similar approaches (A.6 and A.7).

**Strengths And Weaknesses:**

**Strengths**

* The authors carry out an extensive analysis of the behavior of various implicit neural architectures (sinusoidal, gaussian and RFFs) commonly used. Curated examples are provided in support of the theoretical investigation.
* The proposed regularizer is conceptually simple and appears to improve PSNR across evaluations.

**Weaknesses**

* Only images are considered. The paper would be stronger if the proposed regularized was tested on different modalities, such as encoding of audio samples. The experimental evaluation is generally limited - only 500 images from ImageNet (unclear how these are chosen), a subset of STL, and some synthetic data.
* No code is provided. The experimental details provided are not sufficient for reproducibility.


Some minor typos and comments:
* l 162: Fourier transform OF the activation exists
* l 339: Bishop rather than Biship.
* l 528 (appendix), missing reference (??)
* 576 (appendix) erilest -> earliest
* Readability of Lemma A.1 proof could be improved by always specifying the integration domain, particularly the steps where Fubini's theorem is invoked.

---

> ### Author Response · Authors · 2022-08-02
> **Response to Reviewer F1QE Part 1**
>
> We appreciate the thoughtful and positive comments. Please find our answers below:
>
> **1) Only images are considered. The paper would be stronger if the proposed regularized was tested on different modalities, such as encoding of audio samples. The experimental evaluation is generally limited - only 500 images from ImageNet (unclear how these are chosen), a subset of STL, and some synthetic data.**
>
> We considered 1D examples, images, and 3D NeRF-style examples for our experiments since they are commonly used with INRs. Please note that NeRF-style settings are fairly complex and provide a good test bed to advocate the generality and the practical usefulness of the proposed regularization scheme. The reason for using a subset of images from ImageNet is that this is an instance-based optimization setting: that is, a single network has to be trained on each input image. Therefore, evaluating a network over the entire 1M images of the ImageNet is not practical. Further, we believe that since a network is trained per image, the gathered empirical conclusions over a subset can be generalized to other natural images safely. Another example of this practice is the well know work [1]. In there, the size of the image dataset used to evaluate INRs is quite small (e.g. 64 images ). We will release the subset of images that we used for our experiments.
>
> [1] - Tancik, Matthew, et al. "Fourier features let networks learn high frequency functions in low dimensional domains." Advances in Neural Information Processing Systems 33 (2020): 7537-7547.
>
> **2) No code is provided. The experimental details provided are not sufficient for reproducibility.**
>
> We are still working on preparing the code for better usability. We will release our codes soon.
>
> **3) Typos**
>
> We have fixed these in the revised version. Thank you for pointing these out.
>
> **4)  Coordinate-MLPs are finding application even outside of implicit representation of data. It is common to provide "coordinates" as additional inputs to neural network architectures, or to use periodic activation functions (see for example current architectures for diffusion models (1), or neural operators (2, 3) ). I am curious if the authors think the proposed regularized could be helpful even in general deep learning tasks.**
>
> We thank the reviewer for pointing us towards these exciting works. We can indeed safely assume that the proposed regularization has the potential to help many architectures related to these works for converging to better solutions. For instance, Eq. 10 and 11 of our paper are architecture agnostic. That is, the proposed regularization acts as a proxy to ensure that the metric tensor $M$ of the manifold induced by a particular layer (of any architecture) has a lower condition number defined as $\frac{\lambda_{max}}{\lambda_{min}}$, where $\lambda_{max}$ are the maximum and minimum eigenvalues of $M$. This is a desirable property for *many* transformations, since it encourages a smooth behavior of the output with respect to the inputs, *in between the training points*.
>
> However, the extent to which the proposed regularization helps depends on the architecture itself. For instance, ReLU networks (without positional embeddings) are known to converge to smooth solutions implicitly, alleviating the need for an explicit regularization.
>
> Below, we very briefly discuss each of the mentioned works.
>
> *Li et al.*: Their architecture consists of a lifting network $P$, then a series of Fourier layers, and finally, a projection network $Q$. Now, Fourier layers are composed of a Fourier transform, a transformation $R$, an inverse Fourier transformation, and a skip connection with another affine transformation. Fourier transformations are just structured affine/linear transformations, and $R$ is also a linear transformation. Thus, we can conclude that they induce smooth interpolations between training points. Therefore, it all comes down to the properties of the non-linearity $\sigma$ used in each Fourier layer, and the properties of $P$ and $Q$. If $\sigma$ is a ReLU function, and $P, Q$ are typical ReLU-MLPs, then it is reasonable to assume that the proposed regularization might not have a significant effect. On the other hand, if activation functions that can induce wide Fourier spectrums (due to the high Lipschitz constants) such as Sine and Gaussians  are used, then indeed, the proposed regularization can aid in smoothing the function induced by the composite transformation.

---

> ### Author Response · Authors · 2022-08-02
> **Response to Reviewer F1QE Part 2**
>
> *Poli et al.*: They propose to directly optimize operators on the frequency domain, in order to reduce the overhead in converting back and forth between spatial and frequency domains. In summary, their objective is to minimize $||\mathcal{F}(y) - f \circ \mathcal{F}(x) ||$ with respect to an empirical distribution \{ $x,y$ \}, where $\mathcal{F}$ is the Fourier transform and $f$ is a trainable operator in the frequency domain. One can immediately see that the smoothness between the data points of $f \circ \mathcal{F}(x)$ is vital here for a well conditioned estimator. However, the Fourier transformation here is again a linear operator, and thus, is smooth by construction. Therefore, the usefulness of the proposed regularization depends on the properties of $f$. Like in the previous scenario, for instance, if Sine or Gaussians MLPs are used, the proposed regularization can indeed help.
>
>
> *Ho et al.*: With diffusion models, we speculate that the proposed regularization would *not* help. Typically, the time coordinates $t$ fed to the generator $\epsilon(\textbf{x}_t, t)$ is exactly the same in training and inference, and it is not required to generalize to unseen $t$ coordinates. In other words, simply learning the behavior of $\epsilon$ on these specific time coordinates is enough for diffusion models. Therefore, one actually does not need the output of $\epsilon$ to be smooth between the training coordinates along the time dimension.

---

> > ### Comment · Reviewer_F1QE · 2022-08-05
> > **Response to authors**
> >
> > Thank you for the very detailed response. Most of my concerned have been addressed, and I support acceptance of this work.
> >
> > > The reason for using a subset of images from ImageNet is that this is an instance-based optimization setting: that is, a single network has to be trained on each input image.
> >
> > I am aware of the instance-based nature of implicit representation evaluations. My concern is that it is not clear how the subset is selected, and I am sure you can agree with the fact that different subsets could potentially yield different performance rankings. Not all images are created equal: this is the reason for the existence of carefully selected datasets such as the [Kodak Image Dataset](http://www.cs.albany.edu/~xypan/research/snr/Kodak.html) which contain images with different spectral content.
> >
> > > Typically, the time coordinates  fed to the generator  is exactly the same in training and inference, and it is not required to generalize to unseen  coordinates.
> >
> > Unrelated to the paper, but I appreciate the discussion. For continuous-time diffusion models, it can be the case that the model is asked to generalize to unseen $t$, since training is performed by randomly sampling $t$ and computing a denoising loss.

---

> > > ### Author Response · Authors · 2022-08-07
> > > **Response to Reviewer F1QE**
> > >
> > > We thank the reviewer for the enlightening discussion and apologize for misunderstanding the comment regarding the selection of the subset. In the current work, we did not use any specific method to choose the image subset other than random sampling. We agree that a more systematic method that covers different spectral properties could have strengthened the experiments.
> > >
> > > Also, initially, we were unaware of the continuous-time diffusion models. In that case, yes, definitely the proposed technique has the potential to improve the outputs specifically since positional embeddings inherently tend to suppress lower frequencies of the learned function (as we have shown in our paper). We think this indeed is an exciting research question that should be further perused and again, thank the reviewer for bringing these works to our attention.

---

### Official Review · Reviewer_h9m2 · 2022-07-04

**Rating:** 6
**Confidence:** 4
**Soundness:** 3 good
**Presentation:** 4 excellent
**Contribution:** 3 good

**Summary:**

The paper shows that coordinate MLPs lose some of their implicit regularisation properties when they are adapted to better represent high frequency signals. This suggests that implicit regularisation properties of neural networks are not necessarily a consequence of using SGD, but instead related to the network architecture. The authors suggest a new regularisation technique that allows coordinate MLPs to maintain high quality in their representation of low frequency signals, while still allowing improved representation of high frequency signals. The performance of the suggested explicit regularisation is demonstrated by multiple experiments.

**Questions:**

### Q1

In Table1 what is difference of L-PSNR, R-PSNR and T-PSNR? I think I can roughly deduct what they are, but explicitly stating these would be helpful.

### Typos and other minor things
Line 147 typo: positonal

Line 162 typo: transform _of_ the activation?

Figure 2 caption overlaps with x-axis titles

Line 210 typo: follwing

Figure 4 caption typo: densly

Fig 7 caption: is able _to_ achieve


**Limitations:**

The authors comment on which of their results are novel, but the limitations of the proposed regularisation method are not included. It could be useful to include a comment on whether using the proposed method makes training the coordinate MLPs more difficult or requires tuning some training related hyperparameters. Potential societal impact of the work is not discussed, but this doesn't seem very relevant or necessary for this paper.

**Strengths And Weaknesses:**

### Originality

Loss of implicit generalisation in coordinate MLPs has been observed before, but this paper provides some theoretical reasoning to why this happens. The paper also suggests a new method to help resolve this problem.

### Quality

Extensive references to prior work, consistent story throughout the paper, theoretical result of why implicit regularisation of coordinate MLPs deteriorates are mostly supported by proper derivations. For some findings, proper proofs are not provided but intuitive reasoning supported by maths is still given.

### Clarity

Very clear explanations of all relevant concepts, well readable without going into references also for a reader with less expertise in the specific topic. Experiments help to clarify the effects of the proposed method as well as the existing issues with coordinate MLPs. Contributions are very clearly written out.

### Significance

New method proposed with practical use in a field that has recently been popular.

---

> ### Author Response · Authors · 2022-08-02
> **Response for reviewer h9m2**
>
> We are grateful for the valuable comments. Please find our responses below:
>
> **1) In Table 1 what is difference of L-PSNR, R-PSNR and T-PSNR?**
>
> Thank you for pointing out our mistake. Please find the definitions below:
>
> L-PSNR - PSNR evaluated on the left half of the image.
>
> R-PSNR - PSNR evaluated on the right half of the image.
>
> T-PSNR - PSNR evaluated on the total image.
>
> We have included the definitions in the revised paper.
>
> **2) Typos**
>
> We have fixed these in the revised version. Thank you for pointing these out.
>
> **3) The authors comment on which of their results are novel, but the limitations of the proposed regularisation method are not included. It could be useful to include a comment on whether using the proposed method makes training the coordinate MLPs more difficult or requires tuning some training related hyperparameters.**
>
> We have added a limitation section outlining the above.

---

> > ### Comment · Reviewer_h9m2 · 2022-08-08
> > **Response to authors**
> >
> > Thank you for the clarifications and the response, I have no further concerns. I'll keep my score at 6.

---

### Official Review · Reviewer_Vdaq · 2022-07-08

**Rating:** 7
**Confidence:** 3
**Soundness:** 3 good
**Presentation:** 3 good
**Contribution:** 3 good

**Summary:**

This paper studies coordinate-MLPs, a class of multi-layer perceptrons that are trained to parametrize high-frequency signals over a low-dimensional domain.

The paper analyzes three main classes of coordinate-MLP, respectively based on random Fourier features, sinusoidal activations, and Gaussian activations. By proposing a new way of computing the Fourier transform of these MLPs (in the two layers case), the authors observe the effect of the main hyperparameters of the models on the spectrum.
Similarly, the authors study the effect of increasing the depth of the MLPs.

The authors conclude that, in all three cases, changing hyperparameters to allow the model to represent higher frequencies has the effect of decreasing the spectral energy in the lower frequencies.
The authors use this result to propose a regularization scheme based on minimizing the trace of $\mathbf{J}\mathbf{J}^\top$, where $\mathbf{J}$ is the Jacobian of the penultimate layer. This has the effect of limiting the high-frequency components of the MLP associated with low-energy regions of the target signal.

However, the authors also note that one can avoid computing the Jacobian by minimizing a much simpler loss designed to make the output of the penultimate layer smoother along randomly sampled directions.

The paper is concluded with a series of experiments showing that the regularized MLPs can learn to generalize better than their unregularized counterparts, on signals sampled at variable intervals and on signals with specific spectral features (e.g., Fig. 7).

**Questions:**

- In Section 5.2, it is not clear why increasing $a$ means decreasing the energy of the low-frequency components. Can the authors comment on this?

**Limitations:**

The limitations of the work are not really discussed. There's probably something to say about the fact that the theoretical analysis is conducted on 2-layer MLPs and the analysis is assumed to hold on deeper networks.
Also, the authors should comment on the empirical differences between the regularization based on the Jacobian and the simplified regularization.

**Strengths And Weaknesses:**

Overall, I really enjoyed the paper and I can safely recommend its acceptance.
The contributions are entirely novel and the paper is very well-written and explained.
The paper is likely to have a significant impact on the growing community of implicit neural representations/neural fields and gives non-trivial insights into well-known problems.

I apologize that I cannot write a longer review, but I only have a few minor comments on the paper:

- The numbers reported in Table 1 should be explained in the text or the caption.
- The reasoning in lines 252-257 is not immediate to follow. I suggest the authors spend a few more words to facilitate the intuition of the result beyond the empirical observation.
- The comparisons in Figure 4 are difficult to see, I suggest either increasing the size of the images or the resolution (or zooming in on the cutouts).
- Figure 6 could be improved by turning it into parallel line plots instead of scatter plots.
- The authors should comment on the empirical differences between the regularization based on the Jacobian and the simplified regularization. An experiment showing the difference in practice (computational cost, quality of the learned signal, etc.) would be a valuable addition to the paper.
- Typos:
    - Line 47: Biship -> Bishop
    - Line 162: Fourier transform the activation -> Fourier transform of the activation
    - Line 288: Much more simpler -> Much simpler

---

> ### Author Response · Authors · 2022-08-02
> **Response for Reviewer Vdaq**
>
> We thank the reviewer for the insightful and encouraging comments. Please find our responses below.
>
> **1) The numbers reported in Table 1 should be explained in the text or the caption.**
>
> Thank you for pointing this out. The definitions of the metrics are as below:
>
> L-PSNR - PSNR evaluated on the left half of the image.
>
> R-PSNR - PSNR evaluated on the right half of the image.
>
> T-PSNR - PSNR evaluated on the total image.
>
> We have included the definitions in the revised paper.
>
> **2) The reasoning in lines 252-257 is not immediate to follow. I suggest the authors spend a few more words to facilitate the intuition of the result beyond the empirical observation.**
>
> Consider Eq. 9 in the paper:
>
> \begin{equation}
>     min(\kappa'(x)) < \frac{k}{t} < max(\kappa'(x))
> \end{equation}
>
> Eq.7, 8 (in the paper) showed that adding more layers increases the maximum frequency of $\kappa$. This tends to increase the fluctuations of $\kappa$, which in turn increases the magnitude of the first order derivatives $\kappa'$ of $\kappa$. On the other hand, $\beta$ is non-negligible only at points where (1) is satisfied. Note that as $\mathrm{min}(\kappa')$ is increased, for smaller $k$'s, Eq. 9 is not satisfied. In other words, as we add more layers, $\beta$ (in Eq. 7) becomes negligible (smaller) for smaller $k$'s. This, in turn, makes the inner product $<\eta(\cdot), \beta(k,\cdot) >$ smaller for smaller $k$, which means the energy of the smaller frequencies gets reduced (see Eq. 7 in the paper). But a critical assumption of this line of argument is that increasing the frequency of $\kappa$ increases the lower bound of the first order derivatives (min($\kappa'$)), which doesn't always have to be the case. However, our experiments strongly justify this assumption.
>
> We have updated the revised version to explain the above better.
>
> **3) The comparisons in Figure 4 are difficult to see, I suggest either increasing the size of the images or the resolution.**
>
> Thank you. We have increased the Figure size in the revised version.
>
> **4) The authors should comment on the empirical differences between the regularization based on the Jacobian and the simplified regularization.**
>
> We have added a comparison in the Table 5 in the Appendix. Thank you for the suggestion.
>
> **5) Typos**
>
> We have fixed these in the revised version. Thank you for pointing these out.
>
> **6) In Section 5.2, it is not clear why increasing a means decreasing the energy of the low-frequency components. Can the authors comment on this?**
>
> Consider equation 5:
>
> \begin{equation}
>     \sum_{i= 1}^{m} w^{(2)}_i \frac{(2\pi)^{\frac{n}{2}}}{2|\textbf{w}^{(1)}_i|}\delta(\textbf{k}) ( \delta(\frac{\textbf{w}^{(1)}_i}{|\textbf{w}^{(1)}_i|^{2}}\cdot \textbf{k}-a) + \delta(\frac{\textbf{w}^{(1)}_i}{|\textbf{w}^{(1)}_i|^{2}}\cdot \textbf{k}+a))
> \end{equation}
>
> The support of this distribution is localised to those frequencies $\textbf{k}$ for which $\textbf{w}^{(1)}_i \cdot \textbf{k} = \pm a|\textbf{w}^{(1)}_i|^{2}$. Then for positive $a$,
>
> \begin{equation}
>  |\textbf{w}^{(1)}_i \textbf{k}| =  a|\textbf{w}^{(1)}_i|^{2}
>  \leq |\textbf{w}^{(1)}_i||\textbf{k}|.
> \end{equation}
>
> Which gives,
>
> \begin{equation}
>   a|\textbf{w}^{(1)}_i|
>  \leq |\textbf{k}|.
> \end{equation}
>
> Therefore, increasing $a$ increases the lower bound on the frequencies.
>
> **7) The limitations of the work are not really discussed.**
>
> We have added a limitation section.
>
> **8) Also, the authors should comment on the empirical differences between the regularization based on the Jacobian and the simplified regularization.**
>
> We have added a comparison between the regularizers in Appendix. Thank you for the suggestion.

---

> > ### Comment · Reviewer_Vdaq · 2022-08-03
> > **Reply to authors**
> >
> > The authors have addressed all my comments and modified the paper accordingly.
> > My rating remains unchanged, interesting work!

---

### Meta-Review · Area_Chair_YR4h · 2022-08-26

**Recommendation:** Accept
**Confidence:** Certain

**Metareview:**

The paper initially received three positive reviews and a borderline reject one. After the rebuttal, all reviewers are voting for accepting the paper. The area chair agrees with their assessment and follows their recommendation.

**Award:**

No

---

### Decision · Program_Chairs · 2022-09-14

Accept